# The turning and barrier course reveals gait parameters for detecting freezing of gait and measuring the efficacy of deep brain stimulation

**Johanna O'Day**[1,2], **Judy Syrkin-Nikolau**[2¤a], **Chioma Anidi**[2¤b], **Lukasz Kidzinski**[1], **Scott Delp**[1], **Helen Bronte-Stewart**[2,3]*

**1** Department of Bioengineering, Stanford University, Stanford, California, United States of America, **2** Department of Neurology and Neurological Sciences, Stanford University, Stanford, California, United States of America, **3** Department of Neurosurgery, Stanford University, Stanford, California, United States of America

¤a Current address: Cala Health, Burlingame, California, United States of America
¤b Current address: University of Michigan Medical School, Ann Arbor, Michigan, United States of America
* hbs@stanford.edu

**Data Availability Statement:** All gait parameter data files will be available from the public

## Abstract

Freezing of gait (FOG) is a devastating motor symptom of Parkinson's disease that leads to falls, reduced mobility, and decreased quality of life. Reliably eliciting FOG has been difficult in the clinical setting, which has limited discovery of pathophysiology and/or documentation of the efficacy of treatments, such as different frequencies of subthalamic deep brain stimulation (STN DBS). In this study we validated an instrumented gait task, the turning and barrier course (TBC), with the international standard FOG questionnaire question 3 (FOG-Q3, r = 0.74, $p < 0.001$). The TBC is easily assembled and mimics real-life environments that elicit FOG. People with Parkinson's disease who experience FOG (freezers) spent more time freezing during the TBC compared to during forward walking ($p = 0.007$). Freezers also exhibited greater arrhythmicity during non-freezing gait when performing the TBC compared to forward walking ($p = 0.006$); this difference in gait arrhythmicity between tasks was not detected in non-freezers or controls. Freezers' non-freezing gait was more arrhythmic than that of non-freezers or controls during all walking tasks ($p < 0.05$). A logistic regression model determined that a combination of gait arrhythmicity, stride time, shank angular range, and asymmetry had the greatest probability of classifying a step as FOG (area under receiver operating characteristic curve = 0.754). Freezers' percent time freezing and non-freezing gait arrhythmicity decreased, and their shank angular velocity increased in the TBC during both 60 Hz and 140 Hz STN DBS ($p < 0.05$) to non-freezer values. The TBC is a standardized tool for eliciting FOG and demonstrating the efficacy of 60 Hz and 140 Hz STN DBS for gait impairment and FOG. The TBC revealed gait parameters that differentiated freezers from non-freezers and best predicted FOG; these may serve as relevant control variables for closed loop neurostimulation for FOG in Parkinson's disease.

repository SimTK (https://simtk.org/projects/pdwalking, http://doi.org/10.18735/454q-tx50).

**Funding:** This study was supported in part by the Michael J. Fox Foundation (9605, HBS), the National Institute of Neurological Disorders and Stroke (NINDS Grant 5 R21 NS096398-02; HBS, CA, JSN), the Robert and Ruth Halperin Foundation (HBS), the John A. Blume Foundation (HBS), the Helen M. Cahill Award for Research in Parkinson's Disease (HBS), the Stanford Bio-X Bowes Graduate Fellowship (JO), and the National Institutes of Health Big Data to Knowledge (BD2K) Center of Excellence Grant U54EB020405 (SD, JO). Medtronic Inc. provided the devices used in this study but no additional financial support. At the time that the data was collected and study performed, all the authors of the study were employees and funded by Stanford University and the project was funded by the funders listed in the study. Subsequent to this, one of the authors (JSN) became employed by Cala Health. The funders did not have any role in the study design, data collection and analysis, decision to publish, or preparation of the manuscript. Michael J Fox: https://www.michaeljfox.org/ NINDS: https://www.ninds.nih.gov/ Stanford Bio-X: https://biox.stanford.edu/ NIH BD2K: https://commonfund.nih.gov/bd2k.

**Competing interests:** Dr. Helen Bronte-Stewart is a member of a clinical advisory board for Medtronic Inc. and Scott Delp is a scientific advisor and board member of Cala Health, Circuit Therapeutics, and Zebra Medical Technologies, and receives compensation for this service. Dr.Helen Bronte-Stewart and Johanna O'Day have submitted a provisional patent as co-inventors of systems and methods for deep brain stimulation kinematic controllers (patent #S19-551). This does not alter our adherence to PLOS ONE policies on sharing data and materials.

## Introduction

Gait impairment and freezing of gait (FOG) are common in Parkinson's disease, and lead to falls, [1–3] resulting in injury, loss of independence, institutionalization, and even death [4,5]. Over 10 million people are affected by Parkinson's disease (PD) worldwide, and over 80% of people with moderate to advanced PD develop FOG [6]. Gait impairment is characterized by the loss of rhythmic alternating cycles of forward motion of one leg during the stance phase of the other leg, which are represented by the variability of stride time (rhythmicity) and the angular velocity of the lower leg (shank angular velocity) during the swing phase, respectfully. FOG is an intermittent, involuntary inability to perform alternating stepping and usually occurs when patients attempt to initiate walking, turn, or navigate obstacles.

Understanding and treating gait impairment and FOG are paramount unmet needs and were given the highest priority at the National Institute of Neurological Disorders and Stroke 2014 PD conference [7]. Both gait impairment and FOG have unpredictable responses to dopaminergic medication and continuous high frequency open loop subthalamic deep brain stimulation (DBS) [8,9]. Although gait impairment and FOG may improve on continuous lower frequency (60 Hz) DBS, Parkinsonian tremor may worsen, and many patients do not tolerate 60 Hz DBS for long periods of time [10–12]. A closed loop, adaptive system that can adjust stimulation appropriately may be able to improve therapy for FOG and impaired gait. Before this goal can be attained, however, it is important to determine which gait parameters are associated with freezing behavior, which predict freezing events, and the effect of different DBS frequencies on gait impairment and FOG.

Several studies have employed wearable inertial sensors to monitor, detect, and predict FOG using a variety of different gait parameters. The most popular approach has been to use a frequency-based analysis of leg accelerations to capture the "trembling of knees" associated with FOG, and many variations on this approach have been described including the "freeze index" [13] and "Frequency Ratio" [14]. These studies have employed a variety of different FOG-eliciting tasks, such as turning 360 degrees in place for two minutes, walking around cones, or walking during dual tasking [14–22]. These tasks have improved the detection of FOG but are not representative of real-world environments, or cannot objectively measure gait arrhythmicity, which has been correlated with FOG [23–27]. Objective gait measures and standardized gait tasks that reliably induce FOG are needed to study the progression of gait impairment and FOG in PD, and the efficacy of therapeutic interventions.

The goals of this study were to (1) validate a standardized gait task, the turning and barrier course (TBC), which mimics real-life environments and elicits FOG, (2) discover relevant gait parameters for detecting FOG in Parkinson's disease in the TBC, and (3) evaluate the effects of 60 Hz and 140 Hz subthalamic deep brain stimulation (DBS) on quantitative measures of non-freezing gait and FOG.

## Materials and methods

### Human subjects

Twenty-three subjects with PD (8 female), and 12 age-matched healthy controls (11 female), participated in the study. Subjects were recruited from the Stanford Movement Disorders Center and were not pre-selected based on a history of FOG. Subjects were excluded if they had peripheral neuropathy, hip or knee prostheses, structural brain disorders, or any visual or anatomical abnormalities that affected their walking. For all PD subjects, long-acting dopaminergic medication was withdrawn over 24h (72h for extended-release dopamine agonists), and short-acting medication was withdrawn over 12h before all study visits. A certified rater

performed the Unified Parkinson's Disease Rating Scale (UPDRS III) motor disability scale [28], and the Freezing of Gait Questionnaire (FOG-Q, [29]) on all subjects. Four subjects had FOG-Q scores taken from a prior research visit within the last 4 months. Subjects were classified as a freezer or non-freezer based on the FOG-Q question 3 (FOG-Q3): *Do you feel that your feet get glued to the floor while walking, turning or when trying to initiate walking?* The scores were as follows: *0 –never, 1 –about once a month, 2 –about once a week, 3 –about once a day, 4 –whenever walking*. A freezer was defined as a subject who reported a FOG-Q3 $\geq$ 2 or if the subject exhibited a freezing event during the tasks. Control subjects were excluded if they reported neurological deficits or interfering pathology that affected their gait. All subjects gave their written informed consent to participate in the study, which was approved by the FDA and the Stanford Institutional Review Board.

### Experimental protocol

All experiments were performed off therapy (medication and/or DBS). Subjects performed two gait tasks: Forward Walking (FW), which is a standard clinical test of Parkinson's gait, and the TBC, in a randomized order at their self-selected pace. Both tasks started with 20s of quiet standing. For the FW task, subjects walked in a straight line for 10m, turned around and returned, and repeated this for a total of 40 m. We only analyzed data from the straight walking parts of FW. The FW task was conducted in a hallway at least 1.7 m wide formed by a wall and room dividers (Bretford Mobile Screens, Pivot Interiors Inc., Pleasanton, CA). The room dividers were 1.98 m high and a maximum of 1.14 m wide. In the TBC, subjects walked around and through a narrow opening formed by room dividers [25], Fig 1A.

The TBC was enclosed by a row of dividers on one side and a wall on the other, Fig 1B, which limited the subjects' visual field; the aisles of the TBC were the same width as a standard minimum hallway (0.91 meters) in the U.S., and the narrow opening between dividers was the same width as a standard doorway (0.69 meters), Fig 1A. After the initial standing rest period, the subject was instructed to sit on the chair. At the 'Go' command, the subject was instructed to stand up, walk around the dividers twice in an ellipse, and then walk in a 'figure of eight' (i.e., around and through the opening between the dividers), twice, before sitting down again,

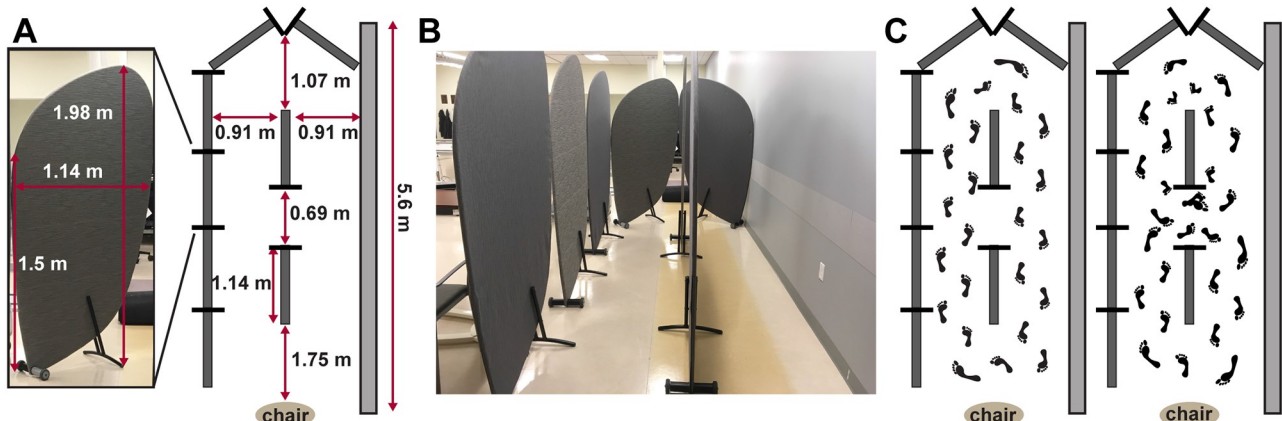

**Fig 1. Turning and Barrier Course (TBC) dimensions and specifications. A**: the individual barrier and course dimensions. Tall barriers limited vision around turns and narrow passageways to simulate a real-world environment. **B**: front view with patient walking in the TBC. **C**: aerial diagram of the TBC with barriers (dark grey bars) and wall (light grey bar). Subjects walked in two ellipses and then two figures of eight around the barriers; this task was repeated starting on both the left and right side, for a total of four ellipses and four figures of eight.

Fig 1C. The subject was then instructed to repeat the task in the opposite direction, for a total of four ellipses and four figures of eight. The direction order was randomized.

## Data acquisition and analysis

Shank angular velocity was measured during the gait tasks using wearable inertial measurement units (IMUs, APDM, Inc., Portland, OR), which were positioned in a standardized manner on the lateral aspect of both shanks. We aligned the IMU on the shank so that the positive Z-axis was directed laterally and measured the angular velocity of the shank in the sagittal plane. Signals from the IMUs' triaxial gyroscope and accelerometer and magnetometer were sampled at 128 Hz. The data were filtered using a zero phase 8th order low pass Butterworth filter with a 9 Hz cut-off frequency, and principal component analysis was used to align the angular velocity with the sagittal plane. Using the sagittal plane angular velocity, the beginning of the swing phase (positive slope zero crossing), end of swing phase (subsequent negative slope zero crossing), and peak shank angular velocities (first positive peak following the beginning of swing phase) were identified, Fig 2.

Forward swing times (time between subsequent zero crossings of the same leg) and stride times (time between consecutive peak angular velocities) were calculated from these data, Fig 2. We used the peaks of the shank angular velocity trace (corresponding to forward swing of the leg) to calculate stride times for each leg to avoid difficulty of discerning heel strikes in PD [30]. These angular velocity peaks were readily identifiable with a computer algorithm and visually. Peaks were marked as steps only if they exceeded a minimum threshold of 10 deg/s, therefore freezing episodes occurred when there was no forward movement of leg or it was below this threshold. Swing angular range was calculated by integrating the sagittal angular velocity curve during the swing time. Swing times and stride times were used to calculate asymmetry and arrhythmicity respectively, during periods when the subject was not freezing. Asymmetry was defined as: asymmetry = $100 \times |\ln(SSWT/LSWT)|$, where SSWT and LSWT correspond to the leg with the shortest and longest mean swing time over the trials, respectively and arrhythmicity was defined as: arrhythmicity = the mean stride time coefficient of variation of both legs [23,26,31]. A large stride time coefficient of variation is indicative of a less rhythmic gait. We developed a "forward freeze index" inspired by the "Freeze Index" [13],

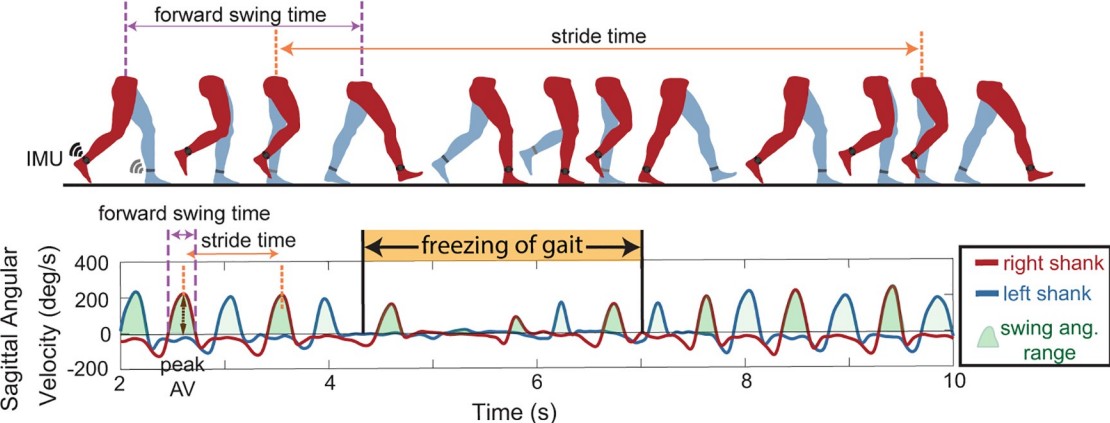

**Fig 2. Gait parameters extracted from inertial measurement units (IMU). Top**: schematic of one gait cycle with IMU on the shank used to define gait parameters including stride time, forward swing time, swing angular range and peak angular velocities (peak AV). **Bottom**: gait parameters extracted from shank sagittal angular velocity data for the left (blue) and right (red) legs during periods of non-freezing walking, and freezing of gait (orange).

and used antero-posterior accelerations instead of vertical accelerations, making it similar to the "Frequency Ratio" [14]. We used a window of 2s rather than 4s because 2s was closer to the mean stride time, and therefore consistent with our other stride-by-stride metrics. The forward freeze index was calculated as the square of the total power in the freeze band (3–8 Hz) over a 2s window, divided by the square of the total power in the locomotor band (0.5–3 Hz) over the same 2s window. External videos of all tasks were acquired on an encrypted clinical iPad (Apple Inc., Sunnyvale, CA) and synchronized with the APDM data capture system through the Videography application (Appologics Inc., Germany).

## A logistic regression model of freezing of gait

We developed a logistic regression model to calculate the probability that a given stride was part of a freezing episode. The model was trained using 8 gait parameters (peak shank angular velocity, stride time, swing angular range, arrhythmicity, asymmetry, forward freeze index, peak shank angular velocity of the previous step, stride time of the previous step) and ground truth binary labels (FOG = 1, no FOG = 0), from an experienced neurologist's (HBS) video-determined ratings of freezing behavior, defined as periods where patient's normal gait pattern changed (usually prior to a freezing episode) and where such behavior ended. VCode software (Hagedorn, Hailpern, & Karahalios, 2008), was used to mark periods of freezing behavior in each video with an accuracy of 10ms. Individual strides were identified using the shank angular velocity trace as described above, and gait parameters were extracted for each stride. The following gait parameters were calculated for each leg independently: peak shank angular velocity, stride time, swing time, and swing angular range. The stride time and peak shank angular velocity were normalized to averages from the subject's FW trial so that subjects could be combined and compared to one another in the model. A step is likely to be a freeze if the step before it has characteristics of a freeze, so the peak shank angular velocity for the previous stride was included as a model input [15]. The swing and stride times for both legs were concatenated to calculate arrhythmicity and asymmetry over the past 6 strides.

Analysis of gait parameters was performed in MATLAB (version 9.2, The MathWorks Inc. Natick, MA, USA), and the logistic regression model was constructed using R (R Core Team (2017)). We used a logistic regression model with a sparse set of features determined by L1 regularization (LASSO) to predict whether a step was freezing or not. To evaluate model performance, we used leave-one-out cross validation (LOOCV), which we refer to as external LOOCV, where we left out a single subject as the test set. We then used the remaining subjects as a training set, and used internal LOOCV, leaving out another subject as an internal test set with which we used L1 regularization (LASSO) to determine a sparse set of features for the model. Regularization minimizes the coefficients of different gait parameters, and the severity to which it does this is determined by the regularization parameter. We found the best regularization parameter (0.01) from the internal training set. This was repeated so that all subjects were left out. We found that the variables selected by the internal LOOCV were consistent across all runs, giving the combination of variables that best identified FOG. In both LOOCVs, we kept subjects, who had multiple visits' worth of data together. For example, if Subject X had two different visits, then data from both visits were either in the training set *or* in the test set.

## Investigating effects of DBS frequency in a subset of the PD cohort

A subset of the cohort, twelve PD subjects (7 freezers and 5 non-freezers), had been treated with at least 21 months of optimized, continuous high frequency subthalamic DBS using an implanted, investigative, concurrent sensing, and stimulating, neurostimulator (Activa® PC + S, FDA-IDE approved; model 3389 leads, Medtronic, Inc.). Kinematic recordings were

obtained, off medication, during randomized presentations of no, 60 Hz, and 140 Hz subthalamic DBS while subjects performed the TBC. The voltage was the same at both frequencies for each subject's subthalamic nucleus. At least five minutes was allotted between experiments to allow the subjects to rest.

## Statistics

A two-way repeated-measures multivariate analysis of variance (MANOVA) test was conducted to assess the effect of Group (Control, Non-Freezer, Freezer) or Task (Forward Walking, TBC ellipse, TBC figure of eight), on average peak shank angular velocity, stride time, asymmetry, and arrhythmicity for the three groups during non-freezing walking while OFF DBS. If a main effect was found in the MANOVA, follow up univariate ANOVAs were used to evaluate significant parameters. Post-hoc pairwise effects were examined using a Bonferroni correction. A three-way repeated measures ANOVA was used to compare the effect of DBS frequency (OFF, 60 Hz, 140 Hz), Group (Non-Freezer, Freezer), or Task (TBC ellipse, TBC figure of eight) during non-freezing walking in the TBC. Post hoc analyses were conducted to compare between stimulation conditions. A Student t-test was used to compare freezers' percent time spent freezing in the TBC ellipses versus TBC figures of eight. Students t-tests were used for the comparison of demographics between the freezer, non-freezer and control groups. A paired samples Wilcoxon test was used to compare UPDRS III scores between visits for subjects with repeated visits. The relationship between percent time freezing and FOG-Q3 response was investigated using a Spearman correlation analysis. The relationship between gait parameters and FOG-Q3 response was investigated using a Spearman correlation analysis. The relationship between percent time freezing and average peak shank angular velocity, stride time, asymmetry, and arrhythmicity during non-freezing walking was investigated using a Pearson correlation analysis to compare freezers' non-freezing walking with the severity of their freezing behavior. All statistical testing was performed in SPSS Version 21, or SigmaPlot (Systat Software, San Jose, CA) using two-tailed tests with significance levels of $p < .05$.

## Results

### Human subjects

Among the 23 PD subjects, there were 8 freezers, 13 non-freezers, and 2 subjects who converted from the definition of a non-freezer to a freezer between two visits. Non-freezers and controls were of similar ages, while freezers were younger (65.9 ± 7.5, 66.9 ± 8.9 years, 57.9 ± 6.14, respectively, $p < 0.05$). Disease duration was similar between the freezer and non-freezer groups (9.3 ± 2.8, 8.9 ± 4.2 years, respectively). Freezers had a higher off medication UPDRS III score than non-freezers (39.8 ± 9.2, 24.1 ± 13.6 respectively, $p < 0.01$), and all PD patients had higher UPDRS III scores than controls ($p < 0.001$). All subjects completed all walking tasks, except two freezers who could not complete the TBC, and one non-freezer whose sensor data was unusable; these three subjects were excluded from the analysis. Three healthy control subjects were excluded due to arthritis (N = 2) or essential tremor (N = 1), which affected their walking. The average total durations of FW and the TBC were 33.1 ± 8.7 and 157.4 ± 88.9 seconds, respectively.

Nine subjects had repeat visits. The length between repeated visits was 430 ± 112 days (mean ± SD) and the repeated visit group's mean UPDRS III score trended higher but was not significant at the second visit (32.4 ± 12.0, 35.7 ± 14.8, respectively, $p = 0.09$). The repeated patient visits were treated independently. Data from 40 visits (9 from controls, 13 from freezers, and 18 from non-freezers) were used to examine how the three different cohorts

completed the gait tasks while OFF stimulation. In assessing the effects of lower and high frequency subthalamic DBS on subjects in the TBC, there were no repeat visits.

## Gait parameters and percent time freezing in the TBC correlated with the FOG-Q3

Subjects' gait arrhythmicity and shank angular velocity during non-freezing gait of the FW task, TBC ellipses and TBC figures of eight were strongly correlated with their self-reported freezing severity (FOG-Q3 score; r = 0.65, 0.46, 0.73 for arrhythmicity respectively, and r = -0.58, -0.46, -0.65 for shank angular velocity respectively, $p < 0.003$ for all), Fig 3. The correlation was strongest during the TBC figures of eight for both gait parameters.

Gait asymmetry was also modestly correlated with FOG-Q3 score in the FW task, the TBC ellipses and the TBC figures of eight (r = 0.44, 0.46, 0.42 respectively, $p < 0.01$ for all). Stride time was not correlated with FOG-Q3 in any of the walking tasks ($p > 0.05$ for all).

During the TBC, all freezers experienced a freezing episode. In total, 217 freezing episodes were identified. Freezers spent more time freezing in the TBC figures of eight than the TBC ellipses (38.23 ± 29.0%, 23.6 0 ± 19.3%, respectively, $p < 0.01$). During FW only one freezer experienced a freezing episode. Freezers spent an average of 33.0 ± 24.2% of the time freezing in the TBC compared to the one freezer who spent 2% of the time freezing during forward walking and was a moderate to severe freezer who spent 59% of the TBC task freezing (as determined by the blinded neurologist). There was a strong correlation between the time spent freezing in the TBC and a subject's report of freezing severity from the FOG-Q3 (r = 0.74, $p < 0.001$), which validates the TBC as a tool for measuring FOG in Parkinson's disease. There was no significant correlation between the time spent freezing during FW and a subject's report of freezing severity from the FOG-Q3 (r = 0.28, $p = 0.075$). These results validate the

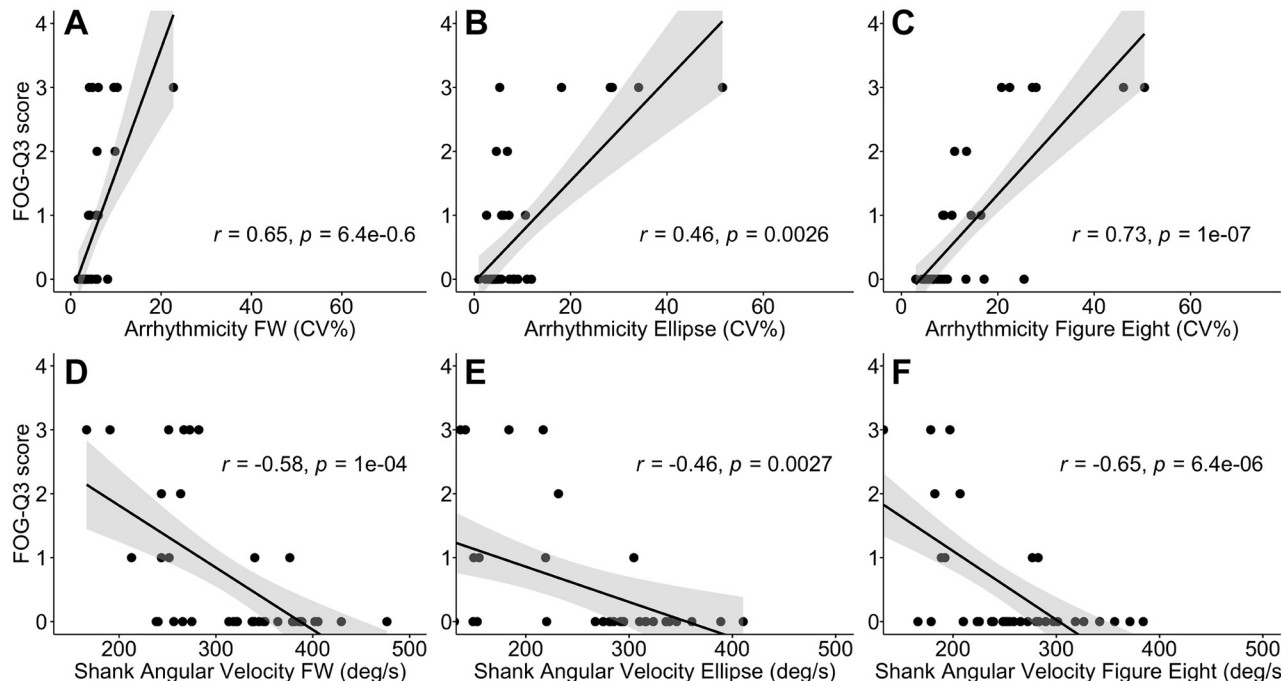

**Fig 3. Relationship between gait parameters and freezing of gait questionnaire question 3 (FOG-Q3) during walking tasks.** Correlation with FOG-Q3 between **A–C**: gait arrhythmicity, and **D–F**: shank angular velocity, during FW **(A, D)**, TBC ellipses **(B, E)**, and TBC figures of eight **(C, F)**. Regression lines (black line) and confidence intervals of the correlation coefficient at 95% (shaded grey), and subjects (black dots) shown.

TBC as a task that can measure gait impairment and FOG; the TBC figures of eight resulted in the strongest correlations between the FOG-Q3 and gait arrhythmicity, shank angular velocity and percent time freezing compared to the TBC ellipses or FW.

### Arrhythmicity during non-freezing gait differentiates freezers from non-freezers

Gait arrhythmicity during non-freezing walking differentiated freezers from non-freezers and from healthy controls in all gait tasks, Fig 4. MANOVA results indicated a main effect of Group (freezer, non-freezer, control, $p < 0.001$) and Task (FW, TBC ellipse, TBC figures of eight, $p < 0.001$), demonstrating that the three groups were distinguishable regardless of task, and the tasks were distinguishable regardless of group. All four of the gait parameters showed significant univariate effects of Group, and all gait parameters except asymmetry showed significant univariate effects of Task. There was an interaction effect of Task*Group ($p = 0.011$), with a univariate effect only in arrhythmicity. Post-hoc pairwise comparisons showed that freezers' non-freezing gait was more arrhythmic than that of non-freezers or controls during all tasks ($p < 0.05$ for all), Fig 4A.

Post-hoc pairwise comparisons showed that freezers' non-freezing gait during both the TBC ellipses and TBC figures of eight demonstrated greater arrhythmicity compared to their non-freezing gait during FW ($p = 0.001$, $p < 0.001$, respectively), and the arrhythmicity of both freezers and non-freezers was greater in the TBC figures of eight than in the TBC ellipses ($p < 0.001$, $p = 0.02$ respectively), Fig 4A. No pairwise effect was detected for non-freezers' or controls' gait arrhythmicity between TBC and FW. There was no Task*Group interaction observed for shank angular velocity, stride time or asymmetry, though the observed power for these variables was low.

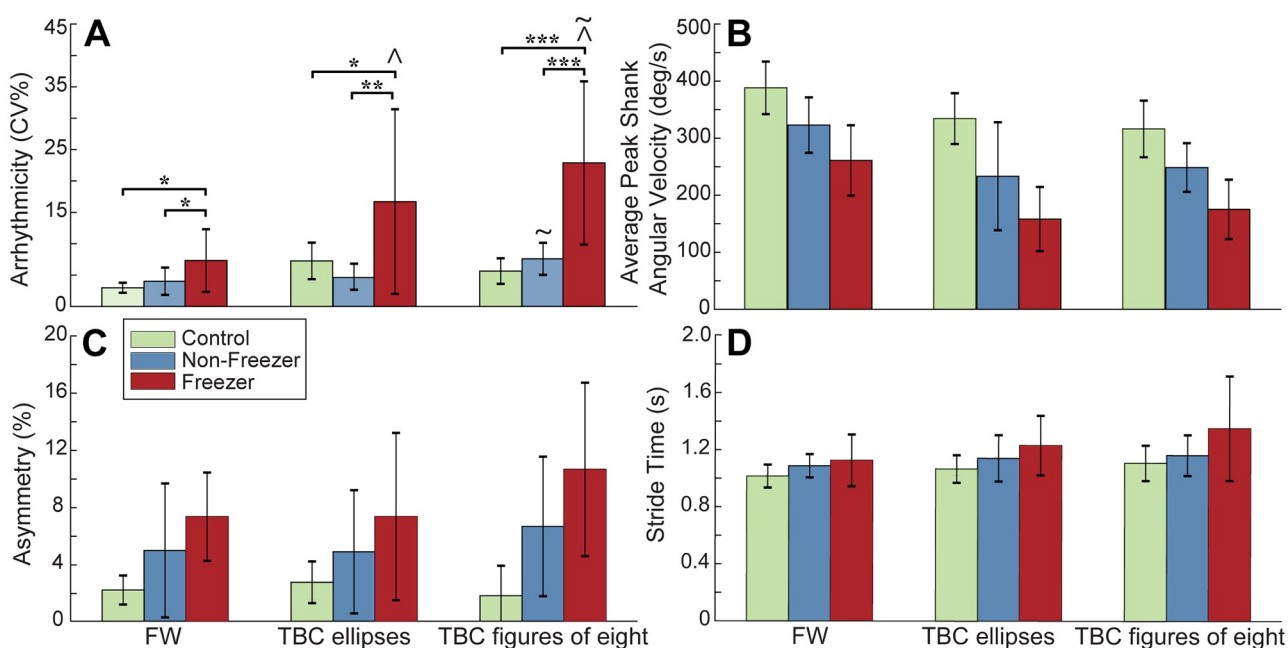

**Fig 4. Group gait parameters during walking tasks. A:** Gait arrhythmicity, **B:** average peak shank angular velocity, **C:** asymmetry, and **D:** stride time in healthy controls, non-freezers and freezers, during non-freezing FW, TBC ellipses and TBC figures of eight. Error bars represent standard deviation. * $p < 0.05$; ** $p < 0.01$; *** $p < 0.001$; ^ $p < 0.05$ TBC ellipses and TBC figures of eight compared to FW in freezers; ~ $p < 0.05$ between TBC ellipses and TBC figures of eight in non-freezers and in freezers.

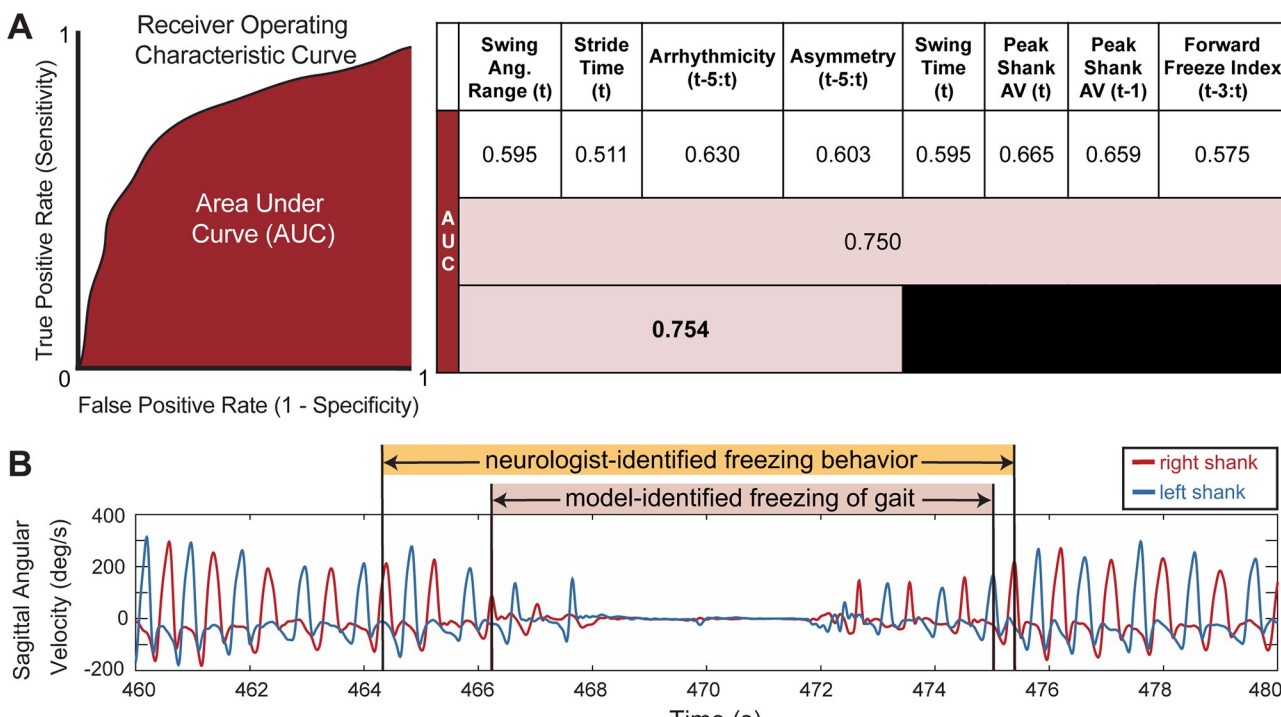

**Fig 5. Logistic regression model performance for different gait parameters. A**: overall model performance: AUC values for different model iterations using leave-one-out cross validation on the freezer group. First row: individual gait parameters; second row: all gait parameters; third row: sparse parameter set chosen from regularization. Peak Shank AV = Peak Shank Angular Velocity. Some metrics are calculated over a window of steps in time: "t-3:t" represents a window from "t-3" or 3 steps earlier, to and including the current step "t". **B**: representative shank angular velocity traces from right and left legs; model-identified freezing events (pink shading) and neurologist-identified freezing behavior (orange shading).

## Gait features in logistic regression model detect freezing on a step-by-step basis

A logistic regression model demonstrated that the best predictor of whether a stride was part of a freezing episode used a combination of four gait parameters: swing angular range, stride time, arrhythmicity, and asymmetry, and had an AUC of 0.754, Fig 5A.

Of these, the gait parameter with the largest coefficient and thereby the strongest predictor of whether a step was part of a freeze, was the arrhythmicity over the last six steps (coefficient of 2.034), followed by stride time (coefficient of 0.0931), swing angular range (coefficient of -0.0615), and finally asymmetry over the last six steps (coefficient of 0.0003), with a model intercept of 0.941. The logistic regression models with single parameters had all coefficients significantly different from zero ($p < 0.001$) but most were only moderately better than chance (AUC = 0.5), first row Fig 5A. A logistic regression model using all gait parameters, second row in Fig 5A, outperformed any single-parameter model but had an AUC (0.750) less than that of the four-parameter-model.

Since the AUC is a threshold-independent assessment of the model, we calculated the accuracy of the model at a threshold of 0.50 (e.g. if the probability that the step was a freeze was over 50% then it was determined to be a freeze). At this threshold, the accuracy of the model to correctly identify a step as freezing or not freezing ((true positives + true negatives)/total number of steps), was 90%. We found that the model often detected a freezing event within the interval defined as freezing behavior by the neurologist, Fig 5B. In this case, the model

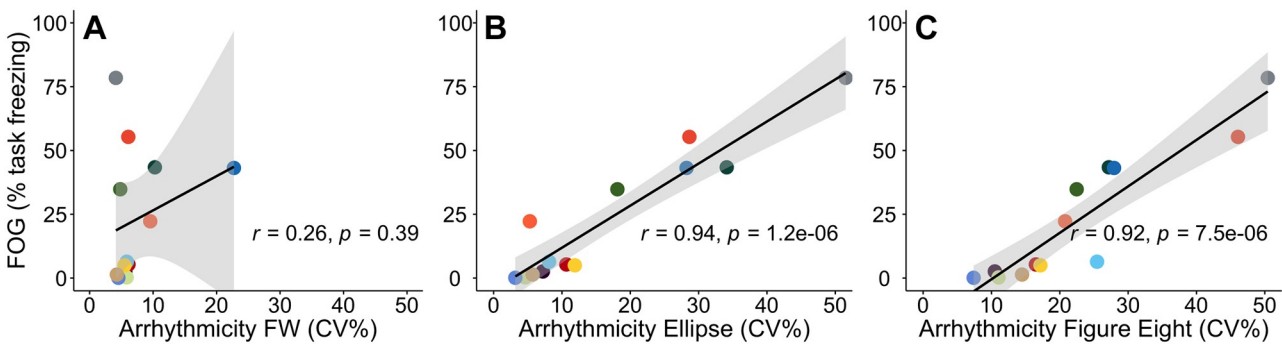

**Fig 6. Relationship between freezers' non-freezing gait arrhythmicity and freezing severity.** Relationship between freezers' non-freezing gait arrhythmicity and percent time freezing **A:** during FW, **B:** during the TBC ellipses, and **C:** during the TBC figures of eight. Regression lines (black line) and confidence intervals of the correlation coefficient at 95% (shaded grey), and subjects (colored dots) shown.

overlapped with the neurologist-identified freezing behavior, though it did not detect some of preceding or succeeding freezing behavior identified by the neurologist. We defined such a case as correct model-identification of a freezing event, and overall, the model correctly identified 77% of the neurologist-identified freezing behavior events, overlapping with neurologist markings within a 2-stride window.

The time spent freezing in the TBC for all subjects, identified by the logistic regression model, correlated with the subject's score on FOG-Q3 ($r = 0.68$, $p < 0.001$). The percent time freezing predicted by the model for the control subjects and non-freezers was less than 1% for each subject, except for one subject who had one step erroneously classified as freezing resulting in 2.5% time spent freezing in the TBC.

### Percent time spent freezing correlated with freezers' gait parameters during non-freezing gait in the TBC

Freezers' gait arrhythmicity during non-freezing gait in both the TBC ellipses and the TBC figures of eight were strongly correlated with their percent time freezing, as determined by the model ($r = 0.94$, $r = 0.92$ respectively, $p < 0.001$ for both), Fig 6. Gait arrhythmicity during FW was not correlated with percent time freezing, Fig 6A.

Freezers' peak shank angular velocity during non-freezing gait in the TBC figures of eight, but not in the TBC ellipses or FW, also correlated with their percent time spent freezing in the TBC ($r = -0.71$, $p < 0.01$, data not shown). There was no correlation between gait asymmetry or stride time during non-freezing walking in the TBC, or between any gait parameter during FW, with percent time freezing in the TBC. These results demonstrated that increased gait arrhythmicity and decreased peak shank angular velocity of non-freezing gait during the TBC were strong markers of FOG severity in PD freezers.

### Sixty Hz and 140 Hz subthalamic DBS improved non-freezing gait impairment and FOG in freezers during the TBC

Gait impairment and FOG improved during both 60 Hz and 140 Hz subthalamic DBS: the percent time spent freezing in the TBC was lower during either 60 Hz or 140 Hz DBS compared to when OFF DBS in freezers ($5 \pm 7\%$, $9 \pm 10\%$, $35 \pm 23\%$, respectively, $p < 0.05$) and was not different from that of non-freezers (whose percent time spent freezing was zero).

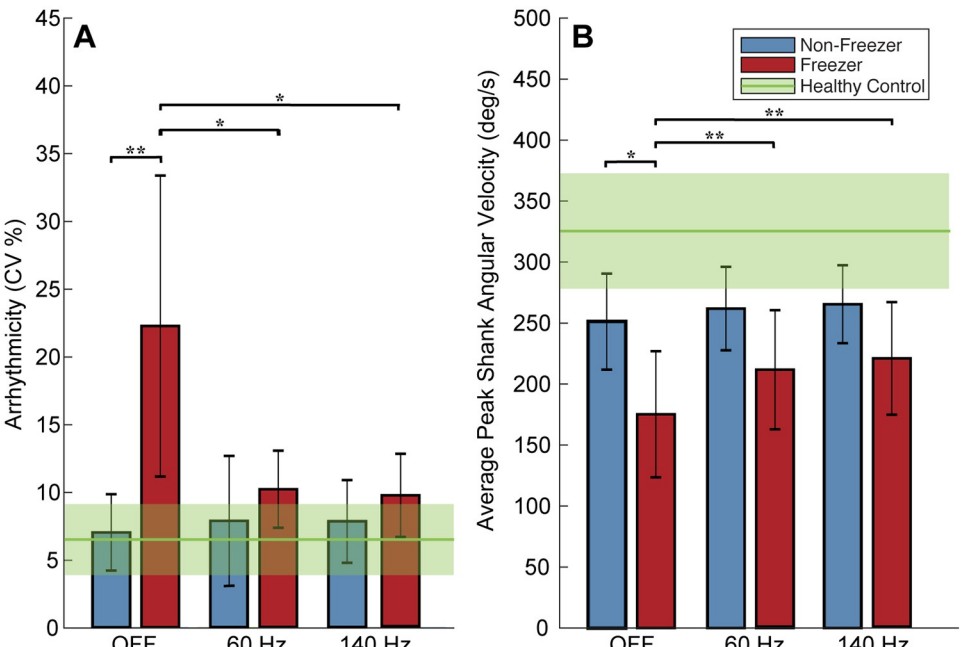

**Fig 7. Gait arrhythmicity and average peak shank angular velocity OFF and during 60 Hz and 140 Hz deep brain stimulation (DBS). A:** Gait arrhythmicity and **B:** average peak shank angular velocity during stimulation conditions. Healthy control averages shown (green line) with standard deviations (shaded green). Error bars represent standard deviation. * denotes $p < 0.05$, ** denotes $p < 0.01$.

Fig 7 demonstrates that there was a statistically significant effect of DBS frequency (OFF, 60 Hz, 140 Hz) on shank angular velocity and arrhythmicity ($p < 0.01$, $p < 0.05$), as well as a statistically significant effect of Task (TBC ellipse, TBC figure of eight) on shank angular velocity ($p < 0.001$) as determined by three-way repeated measures ANOVAs.

Freezers' gait arrhythmicity during the TBC decreased to values not statistically different from those of non-freezers during both 60 Hz and 140 Hz DBS ($p > 0.05$), Fig 7A, despite freezers' arrhythmicity being significantly higher than that of non-freezers OFF DBS ($p < 0.01$). Freezers' shank angular velocity increased during either frequency of DBS ($p < 0.01$), Fig 7B, despite being significantly less than that of non-freezers OFF DBS ($p = 0.036$). There was no effect of DBS on stride time or asymmetry as determined by three-way repeated measures ANOVAs in freezers and DBS had no detectable effect on any of the non-freezers' gait parameters.

OFF DBS freezers had significantly higher arrhythmicity and asymmetry and lower shank angular velocity, than controls ($p < 0.05$ for all), but a similar stride time. OFF DBS there was no difference in non-freezers' arrhythmicity, asymmetry or stride time from those of controls ($p > 0.05$ for all); non-freezers' shank angular velocity was significantly lower than that of controls ($p = 0.031$).

## Discussion

This study has validated the objective measurement of FOG from an instrumented gait task, the turning and barrier course (TBC), with the international standard FOG questionnaire (FOG-Q). The TBC mimicked real-life scenarios that trigger FOG in PD and was superior at eliciting more arrhythmic non-freezing gait, and freezing episodes in freezers compared to 40

meters of forward walking. Freezers' non-freezing gait was more arrhythmic than that of non-freezers or controls irrespective of task.

A logistic regression model demonstrated that a combination of stride time, swing angular range, arrhythmicity, and asymmetry of the past six steps best predicted FOG during the TBC (AUC = 0.754). Freezers' gait arrhythmicity was not only the strongest feature for predicting FOG, but also the non-freezing gait parameter most highly correlated with freezing severity (the percent time freezing).

Freezers' percent time freezing decreased during either 60 Hz or 140 Hz STN DBS and their non-freezing gait arrhythmicity and shank angular velocity was restored to similar values as those of non-freezers.

## The TBC is a validated task for assessing impaired gait and FOG in PD

It has been difficult to develop an objective measure of FOG since it is challenging to elicit FOG in the clinic or laboratory where there are few obstacles, tight corners, or narrow door openings [32]. Tasks that have been shown to provoke FOG include rapid clockwise and counterclockwise 360 degree turns in place [33], in combination with walking through doorways [34], walking with dual tasking [14,35–37], and forward walking tasks including straight walking or turning around cones [38]. We previously validated freezing behavior during a stepping in place task on dual force plates with the FOG-Q3 [23].

In designing the TBC, we desired a forward walking task that included standardized situational triggers for FOG that were representative of real-world scenarios, which could also measure gait parameters such as stride time, swing time, asymmetry and arrhythmicity during both non-freezing and freezing behavior, and gait transitions into and out of freezing [25]. This study validated the TBC with the FOG-Q3, not only with the percent time freezing but also with parameters of gait impairment such as gait arrhythmicity and shank angular velocity that are not available in other tasks. Non-freezing gait arrhythmicity was the most valuable parameter in the validation of the TBC and in differentiating freezers from non-freezers, further supporting the usefulness of the TBC compared to other tasks that cannot measure stride time variability. The TBC was superior to FW in eliciting more arrhythmic gait and FOG events in freezers, and in the correlation of gait arrhythmicity with percent time freezing. This result aligns with previous studies that have shown that freezers exhibit greater arrhythmicity than non-freezers during non-freezing walking or stepping [24,25,31,39,40], though this is the first study to demonstrate this during non-freezing walking and turning to the best of our knowledge. This confirms that the arrhythmicity of non-freezing gait elicited during the TBC is a useful metric to predict the severity of FOG that freezers may experience in the real world, and is a robust measure of freezing behavior even during non-freezing gait.

## A logistic regression model identified freezing events using gait parameters from the TBC

A logistic regression model identified gait arrhythmicity, swing angular range, stride time, and asymmetry as the most important gait parameters for classifying freezing events during the TBC. The model had an AUC of 0.754 and identified the freezing events within the neurologist identified periods of freezing behavior with 77% accuracy. It was interesting that both the neurologist and the model behaved as they were 'trained.' The model's definition of a freezing event was within the neurologist's period, Fig 5B, as the latter identified gait behavior leading up to and after an actual freezing episode, which encompassed complete halts in walking often seen in freezing of gait, but also included gait shuffling, festination, trembling, and shorter strides that often precede and succeed the complete gait arrest. This highlights another variable

in the definition of FOG, some definitions only include 'motor blocks' or events when forward motion stops, whereas others include abnormal freezing behavior in the definition of FOG.

These variable definitions may have contributed to the variation in the accuracy of other IMU-based FOG detection algorithms, which have reported sensitivities and specificities ranging from 73–99% [13–16,18–20,22,41]. Some of these algorithms detected FOG based on high frequency components of leg linear acceleration corresponding to leg trembling-FOG, with lower sensitivity to non-trembling FOG, despite high specificity. The forward freeze index, which measures the relative component of high to low frequency gait components, has been shown to be a useful predictor of FOG in a 360-degree turning task [14]; however this had a lower AUC value in our model compared to other gait parameters, Fig 5A. Explanations for this may include that the TBC task did not include 360 degree turning, which may specifically induce more leg trembling high frequency components of freezing behavior. This supports the clinical experience that FOG manifests with different types of gait impairment depending on what gait task the person with PD is trying to accomplish.

## FOG and gait impairment in freezers improved during STN DBS

We demonstrated that both FOG and predictors of FOG during non-freezing gait improved during 60 Hz and 140 Hz STN DBS while subjects walked in the TBC that mimicked real-life environments that elicit FOG. During the TBC, freezers spent less time freezing when on either frequency of DBS compared to OFF DBS, which is similar to our reports of the effect of DBS on the stepping in place and forward walking tasks [24]. Freezers' gait arrhythmicity also improved on both 60 Hz and 140 Hz DBS, to levels that were not different from that of non-freezers'. Three out of four of non-freezers' gait parameters OFF DBS were not different from those exhibited by healthy controls and all were left unchanged on either frequency of DBS. This 'if it isn't broken, it doesn't need fixing' effect of DBS has been observed in gait [24,42] and in aspects of postural instability [42–44].

Sixty Hz DBS has been shown to be effective in improving axial symptoms in patients with FOG [10,11], though it is not obvious whether 60 Hz versus 140 Hz is better for FOG in real-world walking tasks. Using the clinical assessment of FOG from the MDS-UPDRS III, Ramdhani et al. reported that lower frequency (60 Hz) DBS reduced FOG when high frequency (130 Hz) DBS did not, even shortly after DBS was initiated [45]. Our previous investigations of the effect of 60 Hz and 140 Hz DBS on repetitive stepping in place and on progressive bradykinesia demonstrated that 60 Hz DBS promoted more regularity in ongoing movement, [24,46]

In this study, percent time freezing and gait arrhythmicity improved during either 60 Hz or 140 Hz STN DBS, and to a similar degree. This aligns with a previous report that gait and postural performances with low and high frequency stimulations were largely similar [42], and another demonstrating that 140 Hz STN DBS increased stride length and foot clearing [47], underscoring the increased shank angular velocities demonstrated during STN DBS in this study. Altogether this is valuable assurance for people with PD and clinicians that STN DBS can improve gait and FOG, and that both 60 Hz and 140 Hz improve FOG in real-world walking tasks.

## Limitations

Our logistic regression model utilized data from only one IMU from a small cohort of freezers. Although this resulted in interpretable gait features and an accuracy within that of several other FOG models, it could be improved. Multiple IMUs on different parts of the body may add sensitivity. The model, being a binary classifier, attempted to capture all of the variability in freezing behavior with just two labels: "FOG" or "not FOG". A different model might use

multiple classes, where the classifier discriminates between unimpaired walking, a completely halted gait freeze event, shuffling, and a start hesitation. In addition, only freezers were used to train and test the logistic regression model, so that the incidence of freezing events was sufficient. Future models might include bootstrapping methods, evaluate the data from multiple IMUs, or more data to increase the sizes of the training and test sets.

## Conclusions

Tools and tasks such as the instrumented TBC are necessary for designing and assessing personalized interventions and therapies for gait impairment and FOG in PD. We have validated and demonstrated the utility of the instrumented TBC for eliciting FOG, for revealing gait parameters that identify freezers and predict FOG during non-freezing gait, and for measuring the efficacy of different frequencies of STN DBS. From the TBC experimental data and a logistic regression model, we have identified the gait parameters that are most likely to predict freezing events and which may be useful in closed loop DBS for gait impairment and FOG.

## Acknowledgments

We thank Matthew Petrucci, Tom Prieto, Jordan Parker, Varsha Prabhakar, Raumin Neuville, Ross Anderson, and Amaris Martinez for their support during the experiments and helpful comments. We would also like to thank our dedicated patient population who contributed their time to participating in our study (ClinicalTrials.gov Identifier: NCT02304848).

## Author Contributions

**Conceptualization:** Johanna O'Day, Judy Syrkin-Nikolau, Helen Bronte-Stewart.

**Data curation:** Johanna O'Day, Judy Syrkin-Nikolau, Chioma Anidi.

**Formal analysis:** Johanna O'Day, Lukasz Kidzinski, Scott Delp.

**Funding acquisition:** Scott Delp, Helen Bronte-Stewart.

**Investigation:** Johanna O'Day, Helen Bronte-Stewart.

**Methodology:** Johanna O'Day, Judy Syrkin-Nikolau, Chioma Anidi, Helen Bronte-Stewart.

**Project administration:** Judy Syrkin-Nikolau, Helen Bronte-Stewart.

**Resources:** Helen Bronte-Stewart.

**Software:** Lukasz Kidzinski.

**Supervision:** Scott Delp, Helen Bronte-Stewart.

**Visualization:** Johanna O'Day, Lukasz Kidzinski.

**Writing – original draft:** Johanna O'Day.

**Writing – review & editing:** Johanna O'Day, Judy Syrkin-Nikolau, Chioma Anidi, Lukasz Kidzinski, Scott Delp, Helen Bronte-Stewart.

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
