## [Decision Letter · Decision Letter 0]

11 Feb 2020

PONE-D-19-35218

The turning and barrier course reveals gait parameters for detecting freezing of gait and measuring the efficacy of deep brain stimulation

PLOS ONE

Dear Dr Bronte-Stewart,

Thank you for submitting your manuscript to PLOS ONE. After careful consideration, we feel that it has merit but does not fully meet PLOS ONE’s publication criteria as it currently stands. Therefore, we invite you to submit a revised version of the manuscript that addresses the points raised during the review process.

Thanks for the submission. Both reviewers indicated that the paper is suitable for publication in the journal, but it needs carefully reviewed. Please be attentive in the comments and improve the manuscript.

We would appreciate receiving your revised manuscript by Mar 23 2020 11:59PM. To enhance the reproducibility of your results, we recommend that if applicable you deposit your laboratory protocols in protocols.io, where a protocol can be assigned its own identifier (DOI) such that it can be cited independently in the future. For instructions see: http://journals.plos.org/plosone/s/submission-guidelines#loc-laboratory-protocols

We look forward to receiving your revised manuscript.

Kind regards,

Fabio A. Barbieri, PhD

Academic Editor

PLOS ONE

Journal Requirements:

"This study was supported in part by the Michael J. Fox Foundation (9605, HBS), the National Institute of Neurological Disorders and Stroke (NINDS Grant 5 R21 NS096398-02; HBS, CA, JSN), the Robert and Ruth Halperin Foundation (HBS), the John A. Blume Foundation (HBS), the Helen M. Cahill Award for Research in Parkinson's Disease (HBS), the Stanford Bio-X Bowes Graduate Fellowship (JO), and the National Institutes of Health Big Data to Knowledge (BD2K) Center of Excellence Grant U54EB020405 (SD, JO). Medtronic Inc. provided the devices used in this study but no additional financial support. The funders had no role in study design, data collection and analysis, decision to publish, or preparation of the manuscript.

Michael J Fox: https://www.michaeljfox.org/

NINDS: https://www.ninds.nih.gov/

Stanford Bio-X: https://biox.stanford.edu/

NIH BD2K: " ext-link-type="uri" xlink:type="simple">https://commonfund.nih.gov/bd2k"

We note that one or more of the authors are employed by a commercial company: Cala Health.

"I have read the journal's policy and the authors of this manuscript have the following competing interests: Dr. Helen Bronte-Stewart is a member of a clinical advisory board for Medtronic Inc." 

Reviewers' comments:

Reviewer's Responses to Questions

**Comments to the Author**

1. Is the manuscript technically sound, and do the data support the conclusions?

Reviewer #1: Yes

Reviewer #2: Partly

2. Has the statistical analysis been performed appropriately and rigorously? 

Reviewer #1: Yes

Reviewer #2: Yes

3. Have the authors made all data underlying the findings in their manuscript fully available?

Reviewer #1: Yes

Reviewer #2: No

4. Is the manuscript presented in an intelligible fashion and written in standard English?

Reviewer #1: Yes

Reviewer #2: Yes

5. Review Comments to the Author

Reviewer #1: Comments on The turning and barrier course reveals gait parameters for detecting freezing of gait and measuring the efficacy of deep brain stimulation

This represents an interesting piece of scientific work, clearly and accurately written, presenting new tasks and protocol to induce FoG events in a clinic-like setup. The authors argue that “TBC is easily assembled and mimics real-life environments that elicit FOG”. However, it requires some room and specific equipment to be applied. The continuous turning task, in comparison, has shown to be very sensitive in detecting freezing events while not requiring either room or special equipment. As the current results have clear potential applicability in clinical settings, it would be desirable to make a comparison between the evaluation tasks, discussing potential benefits of using TBC rather than the turning task. I understand that such an argumentation should go beyond the point that the task “mimics real-life environments”, using the presented data on gait analysis as the main point.

Lines 227-8: “students t-test” should be “Student t-test”.

Lines 229-30: “Paired t-tests were used to compare UPDRS III 230 scores between visits for subjects with repeated visits.” Due to the nature of these data, a corresponding non-parametric test should be used.

Lines 507-11: The authors stated that the normalization procedure allowed comparison among individuals with different natural walking speeds. In the sequence, the authors propose that in future studies absolute measures could be tested. From my perspective, (a) speed variability is a feature of this group, and (b) speed normalization is an appropriate procedure for analysis, and so any of these points could not be considered as limitations. The suggestion of using absolute speed is contradictory to the reasons leading to data normalization.

Reviewer #2: The goals of this study were to (1) validate a standardized gait task, the turning and barrier course (TBC), which mimics real-life environments and elicits FOG, (2) discover relevant gait parameters for detecting FOG in Parkinson’s disease in the TBC, and (3) evaluate the effects of 60 Hz and 140 Hz subthalamic deep brain stimulation (DBS) on quantitative measures of non freezing gait and FOG. The paper is interesting, but it needs to have some questions of data analysis answered to support the conclusions.

Insert a reference for the method of identifying the phases of the gait from the angular velocity of the shank. Is this method validated to identify gait cycles in patients with Parkinson's disease, especially with freezing episodes? The calculation of the time of the phases is important because it will influence the logistic regression model.

How did the freezing episodes affect the identification of the gait cycle? Figure 4B shows the difficulty of identifying gait cycles during freezing episodes using the shank's angular velocity.

Detail the reasons for changing the way of calculating the Freeze index.

Why was it analyzed about the rectilinear movement in the Forward Walking task? There is a difference between the tasks because the others have curves.

A figure would be needed showing the correlations between the variables and the FOG-Q3.

Figure 3B could present the values of peak angular velocity shank that was used for the statistic

Please check the correlation between Arrhythmicity and FOG (Figure 5ª). I don't think there is a correlation given the dispersion of individual data.

The analysis of repeated visits is confusing. What are these subjects?

6. PLOS authors have the option to publish the peer review history of their article (what does this mean?). If published, this will include your full peer review and any attached files.

Reviewer #1: Yes: Luis Augusto Teixeira

Reviewer #2: Yes: Daniel Boari Coelho

---

## [Author Response · Author response to Decision Letter 0]

20 Mar 2020

We greatly appreciate these reviews, we really think they have improved the quality of the manuscript and we appreciated the reviewers’ time and energy into their reviews. Here we address each reviewer’s suggestions and questions, and we outline where the changes have been made in the revised manuscript. 

Reviewer #1: 

(1) This represents an interesting piece of scientific work, clearly and accurately written, presenting new tasks and protocol to induce FoG events in a clinic-like setup. The authors argue that “TBC is easily assembled and mimics real-life environments that elicit FOG”. However, it requires some room and specific equipment to be applied. The continuous turning task, in comparison, has shown to be very sensitive in detecting freezing events while not requiring either room or special equipment. As the current results have clear potential applicability in clinical settings, it would be desirable to make a comparison between the evaluation tasks, discussing potential benefits of using TBC rather than the turning task. I understand that such an argumentation should go beyond the point that the task “mimics real-life environments”, using the presented data on gait analysis as the main point.

Response: We have included in the discussion, a more in-depth comparison between the evaluation tasks (page 20, line 452-460). The TBC also allows for calculation of additional metrics such as stride time, swing time, asymmetry and arrhythmicity during both non-freezing and freezing behavior in a real-life environment. In addition, the TBC allows measurement of gait transitions into and out of freezing. 

(2) Lines 227-8: “students t-test” should be “Student t-test.”

Response: This has been addressed in the revised manuscript (page 11, line 234).

(3) Lines 229-30: “Paired t-tests were used to compare UPDRS III 230 scores between visits for subjects with repeated visits.” Due to the nature of these data, a corresponding non-parametric test should be used.

Response: We used a non-parametric test, a paired samples Wilcoxon test and the difference was not significant (p = 0.09). We have adjusted the manuscript to address this change (page 11, line 236-7) and (page 12, lines 262-3).

(4) Lines 507-11: The authors stated that the normalization procedure allowed comparison among individuals with different natural walking speeds. In the sequence, the authors propose that in future studies absolute measures could be tested. From my perspective, (a) speed variability is a feature of this group, and (b) speed normalization is an appropriate procedure for analysis, and so any of these points could not be considered as limitations. The suggestion of using absolute speed is contradictory to the reasons leading to data normalization.

Response: We agree and appreciate this being brought to our attention, lines (507-11) have been removed to alleviate this contradiction.

Reviewer #2: 

The goals of this study were to (1) validate a standardized gait task, the turning and barrier course (TBC), which mimics real-life environments and elicits FOG, (2) discover relevant gait parameters for detecting FOG in Parkinson’s disease in the TBC, and (3) evaluate the effects of 60 Hz and 140 Hz subthalamic deep brain stimulation (DBS) on quantitative measures of non freezing gait and FOG. The paper is interesting, but it needs to have some questions of data analysis answered to support the conclusions.

(1) Insert a reference for the method of identifying the phases of the gait from the angular velocity of the shank. Is this method validated to identify gait cycles in patients with Parkinson's disease, especially with freezing episodes? The calculation of the time of the phases is important because it will influence the logistic regression model.

Response: We agree that the calculation of the time of the gait phases is important. We used a validated method to identify gait cycles (Salarian et al., 2004), and we have clarified and inserted that reference in the manuscript (page 7-8, line 156-61). We have used this method in two other publications (Syrkin-Nikolau et al., 2017, Anidi et al., 2018).

(2) How did the freezing episodes affect the identification of the gait cycle? Figure 4B shows the difficulty of identifying gait cycles during freezing episodes using the shank's angular velocity

.

Response: Thank you; we agree and we did not use the standard heel strike to identify the start of a gait cycle for this reason. We used the peak of the angular velocity trace because this was a readily identifiable peak, both with a computer algorithm and visually. Angular velocity peaks were marked as steps only if they exceeded a minimum threshold of 10 deg/s. Therefore, during complete cessation of gait, steps were not identified. It was precisely for this reason, that we identified freezing episodes as a lack of leg swinging, rather than from heel strikes. We have clarified this in the manuscript (pg. 8 line 159-61).

(3) Detail the reasons for changing the way of calculating the Freeze index.

Response: We adapted the freeze index window to be more consistent with our stride by stride metrics which were on average 1-2 seconds rather than 4 seconds. Please see page 8, line 170-72.

(4) Why was it analyzed about the rectilinear movement in the Forward Walking task? There is a difference between the tasks because the others have curves.

Response: The forward walking task was used as a control task that does not typically elicit freezing. In that task we only analyzed rectilinear walking as it reflects the way that patients are usually assessed in clinics. We wanted to compare how well forward walking elicited freezing behavior compared to the turning and barrier course because it is known that freezing is not often elicited during the clinical assessment of gait. This is a large reason why we developed the TBC because of difficulty of eliciting FOG in forward walking.

(5) A figure would be needed showing the correlations between the variables and the FOG-Q3.

Figure 3B could present the values of peak angular velocity shank that was used for the statistic

Response: We thank the reviewer for this insight and we found that gait arrhythmicity and shank angular velocity did correlate with the FOGQ-3 in addition to the percent time freezing. We have added a figure, Figure 3, and have updated the manuscript (page 12-13, line 269-86).

(6) Please check the correlation between Arrhythmicity and FOG (Figure 5ª). I don't think there is a correlation given the dispersion of individual data.

Response: The reviewer is correct, there was no significant correlation between percent time freezing and forward walking (Figure 5a – now Figure 6a). We have made this clearer in the manuscript and in the figure (page 17, line 378-9).

(7) The analysis of repeated visits is confusing. What are these subjects?

Response: We had subjects with repeated visits that were, on average, over year apart, 430 ± 112 days (mean ± SD), so we included them as separate visits.

(8) PLOS authors have the option to publish the peer review history of their article (what does this mean?). If published, this will include your full peer review and any attached files.

Response: Yes

References

Anidi C, O’Day JJ, Anderson RW, Afzal MF, Syrkin-Nikolau J, Velisar A, et al. Neuromodulation targets pathological not physiological beta bursts during gait in Parkinson’s disease. Neurobiol Dis. 2018;120.

Salarian A, Russmann H, Vingerhoets FJG, Dehollain C, Blanc Y, Burkhard PR, et al. Gait assessment in Parkinson’s disease: Toward an ambulatory system for long-term monitoring. IEEE Trans Biomed Eng. 2004;51(8):1434–43.

Syrkin-Nikolau J, Koop MM, Prieto T, Anidi C, Afzal MF, Velisar A, et al. Subthalamic neural entropy is a feature of freezing of gait in freely moving people with Parkinson’s disease. Neurobiol Dis [Internet]. 2017;108(June):288–97. Available from: http://dx.doi.org/10.1016/j.nbd.2017.09.002

---

## [Decision Letter · Decision Letter 1]

6 Apr 2020

The turning and barrier course reveals gait parameters for detecting freezing of gait and measuring the efficacy of deep brain stimulation

PONE-D-19-35218R1

Dear Dr. Bronte-Stewart,

We are pleased to inform you that your manuscript has been judged scientifically suitable for publication and will be formally accepted for publication once it complies with all outstanding technical requirements.

With kind regards,

Fabio A. Barbieri, PhD

Academic Editor

PLOS ONE

Additional Editor Comments (optional):

Reviewers' comments:

Reviewer's Responses to Questions

**Comments to the Author**

1. If the authors have adequately addressed your comments raised in a previous round of review and you feel that this manuscript is now acceptable for publication, you may indicate that here to bypass the “Comments to the Author” section, enter your conflict of interest statement in the “Confidential to Editor” section, and submit your "Accept" recommendation.

Reviewer #1: All comments have been addressed

Reviewer #2: All comments have been addressed

2. Is the manuscript technically sound, and do the data support the conclusions?

Reviewer #1: Yes

Reviewer #2: Yes

3. Has the statistical analysis been performed appropriately and rigorously? 

Reviewer #1: Yes

Reviewer #2: Yes

4. Have the authors made all data underlying the findings in their manuscript fully available?

Reviewer #1: Yes

Reviewer #2: Yes

5. Is the manuscript presented in an intelligible fashion and written in standard English?

Reviewer #1: Yes

Reviewer #2: Yes

6. Review Comments to the Author

Reviewer #1: (No Response)

Reviewer #2: All comments have been addressed. There was an improvement in the description of the methods and results.

7. PLOS authors have the option to publish the peer review history of their article (what does this mean?). If published, this will include your full peer review and any attached files.

Reviewer #1: Yes: Luis Augusto Teixeira

Reviewer #2: Yes: Daniel Boari Coelho

---

## [Editor Report · Acceptance letter]

20 Apr 2020

PONE-D-19-35218R1 

The turning and barrier course reveals gait parameters for detecting freezing of gait and measuring the efficacy of deep brain stimulation 

Dear Dr. Bronte-Stewart:

I am pleased to inform you that your manuscript has been deemed suitable for publication in PLOS ONE. Congratulations! Your manuscript is now with our production department. 

With kind regards,

on behalf of

Dr. Fabio A. Barbieri 

Academic Editor

PLOS ONE